# Experimental Approach for Printability Assessment: Toward a Practical Decision-Making Framework of Printability for Cementitious Materials

**Zoubeir Lafhaj \*, Andry Zaid Rabenantoandro, Soufiane el Moussaoui, Zakaria Dakhli and Nicolas Youssef**

Centrale Lille, Civil Engineering, Cité Scientifique, 59651 Villeneuve-d'Ascq, France;
andry-zaid.rabenantoandro@centralelille.fr (A.Z.R.); soufiane.el-moussaoui@centralelille.fr (S.e.M.);
zakaria.dakhli@centralelille.fr (Z.D.); nicolas.youssef@centralelille.fr (N.Y.)

\* Correspondence: zoubeir.lafhaj@centralelille.fr

**Abstract:** The objective of this paper is to propose a pre-experimental framework of printability pre-assessment of cementitious materials. This study firstly presents a general review of additive manufacturing in construction and then examines the main characteristic of the material formulation and printability properties based on extrusion technique. This framework comes with experimental tests to determine a qualitative printability index of mixtures. It uses mix-designs reported in the literature to define interval ratio of mixture design to be investigated in this study. The focus was put on two criteria that influence the formulation namely flowability and buildability. Two practiced based tests, mini slump and cone penetrometer, were used to draw the flowability and buildability dimensionless index. The results were analyzed by introducing an optimal printability coefficient and examining their time evolution. An optimal time of printing was determined Toptimal. Finally, a 3D mortar printing system and its operational process are presented. Then, based on the measurement, the optimal mixture is identified and printed in a large-scale geometry.

**Keywords:** 3D printing; extrusion construction industry; additive manufacturing; fresh mortar; mortar printing; measurement; printability; flowability; buildability

## 1. Introduction

Manufacturing has become the backbone economy in the world. In the European economy, it has flourished with 9.2% of enterprises in the member states were classified as manufacturing in 2012 [1]. Along with new manufacturing methods, industry 4.0 was describe as the trend of using these new automation and manufacturing technologies in the environments of construction [2]. A new process of manufacturing known as 3D printing or additive manufacturing (AM) has emerged in the 1980s. AM is known as the process of joining materials to make objects from three-dimensional (3D) model data, usually layer by layer, as opposed to subtractive manufacturing methodologies [3]. It can transform a virtual model into a real structure, which is already the case in the manufacturing industry: aeronautics, automotive, aerospace, medical and pharmaceutical [4]. AM can bring significant improvement in the fields of construction in terms of safety, economic benefits and gain in time of completion [5–7].

Excessive energy consumption, harmful emissions, inefficiency of traditional work, low productivity, huge production of waste and limited design are all the main concerns that have led to introduce 3D printing in the construction industry [8]. This technique offers valuable advantages: high-speed construction, removal of formwork, less heavy labor, topological optimization, unrecognized mechanical properties of the structure and above all a great design flexibility [8–13].

Today, the construction sector is being digitalized, an operation that has invaded the design phase and begins to reach the execution phase [14]. A Boston Consulting Group (BCG) study in 2018 confirms that companies around the world are ready to use this technology and take advantage of its potential value [15]. However, this technology remains under-development, due to scaling up difficulties, absence of a business model in Europe, finished product quality and the intervention of several disciplines: design, architecture, materials science, computer and robotics [16].

In the past 20 years, the development of niche research on 3D printing in construction and automation flourished. In the international, here are a non-exhaustive name such as Contour Crafting, 3D-P (MIT) in United State, Winsun in China, D-Shape from Italy, CyBe Construction from the Netherlands, COBOD International from Danemark, Vertico from the Netherlands, Apis COR from Russia and XtreeE from France [17–19]. These niches have flourished but there are still major issues in the regulation of the material used in 3D printing for construction. In the European context various efforts have been made to facilitate introduction of new material linking normative agencies like Scientific and Technical Centre for Building (STCB) to startups or universities. The main country where the technology is sold for demonstrations purpose is in the Middle East region (Saudi Arabia, Qatar and Dubai) due to their investment in innovation policies.

The core of digital construction and 3D mortar printing is the processing of new cementitious materials [13]. Extrusion based techniques is the most used techniques of 3D printing in construction. Typically, a gantry or robotic arm is used for positioning, a pump is used for material distribution and the hoses is then connected to a nozzle as the printer head [20]. In the distribution phase, the material is required to be easily pumpable [21]. In the deposition phase, the material is required to be able to withstand his own weight and be stacked stably for the other layers to come [22].

However, it is commonly accepted that the material must respond to a number of specifications in order to be printable. The key challenges for printability were outlined by Wangler et al. and also by Le et al. [12,23]. The focus of those researches was to assess the fresh properties of printable mortar. The mortar fresh properties of material using extrusion technique presented and assessed. Cementitious materials for extrusion-based mortar printing must exhibit paradox rheological properties to comply with the required criteria [24]. The printability of the material used is still the major challenge in the field [25]. There is no relevant guideline for the 3D printing in construction in terms of material formulation or machinery specification. Especially there is no normative standard to follow in order to formulate a ready to uses material for AM. Thus, various efforts must be made for standardizing the technology in the construction industry [26,27].

Existing study on material printability focus on a yield stress-based mixture design approach, or numerical approach for 3D printable mortars [28,29]. This study will evaluate the materials performance of different printing trials. A front-end experimental framework of printability for cementitious materials is proposed long with the material and methodology used to confirm the framework. It proposed an overview of the literature review on a practice-based test to give a first assessment of a printable material. It examines the intrinsic properties of fresh mortar to introduce printability coefficients and their evolution during time and the 3D printing process.

A qualitative printability evaluation was set, and the formulations compared according to those indicators. In this study only the fresh properties of the printing mixture are considered, further research is under investigation for the hardened properties of printing mixtures. The development of this kind of framework especially dedicated to laboratory testing will foster 3D printing evolution and contribute to establish guidelines.

## 2. State of the Art and Research Objectives

Through the literature, researchers have proposed different criteria to assess the fresh properties of printable mortar. Most common criteria presented in the literature are extrudability, pumpability buildability, open time, flowability/workability and interlayer adhesion [30–33].

Le et al. [23] proposed extrudability, workability, open time and buildability as essential criteria to identify printability characteristic of mortar. Each of these criteria was determined using a different test. The ability to extrude was determined by extruding filament of 300 mm length and visually checking for blockages and fractures through a 9 mm diameter nozzle. Workability was determined by a two-point test method to characterize Bingham fluid behavior in terms of shear strength. Open time was estimated by determining the slump loss using a mini cone apparatus [23,34].

Kazemian et al. proposed a framework for laboratory testing of printing mortar in a fresh state. The procedure was designed based on the properties of printed layers, these properties was then related to the employed pumping or extrusion mechanism [35]. The initial step of the framework is to design an initial mixture, second step test the print quality, third step shape stability, fourth step printability windows and finally a full-size test.

Papachristoforou et al. proposed an evaluation of workability parameters of fresh mortar used in 3D printing in construction. An experimental measurement was done composed by four tests: flow table for workability, International Center for Aggregate Research (ICAR) rheometer for rheological parameter, Vicat for the setting parameter and an applied electronical experimentation based on measuring the power consumption of the rotating screw extruder [36].

Biranchi et al. proposed a framework of experiment to evaluate mix proportion variation depending on extrudability, shape retention, buildability and thixotropic open time (TOT). Those properties of material are critical at early-age in order to characterize the 3D printable geopolymer [37].

Nerella et al. focused on proposing a methodology for characterizing the extrudability of cement-based materials for digital construction, both quantitatively and inline. The energy consumed per extruded unit volume was defined as a UEE (unit extrusion energy). The UEE was used to assess the extrudability and was compared with a simple ram-extruder, slump-flow and viscometer tests.

Other study such as Wolfs et al., Hosseini et al. and Marchment et al. focused on the bond between layers [38–41]. As previously reported in the literature, the interlayer adhesion between two printed layers is time sensitive [38].

## 2.1. Literature Survey of 3D Printable Mixture

Table 1 presents the common criteria for printability [23]. The printability of a possible printable material must respond to these criteria extrudability, buildability, open time, flowability/workability/pumpability and interlayer adhesion. Each criterion is described and presented with it control dispositive. A proposed measurement test is presented that regroups different literature surveys [23,31,37,42–45].

Table 2 illustrates the different printable mixtures as shown in the literature. Portland cement (PC) is the prime ingredient binder in most of the formulations presented through the literature due to its inherent thixotropic property [46]. For example, the ratio of water to cement occupies a wide range of values from 0.27 to 0.51 and no limitations can be concluded. The aggregates volume for cementitious material such as concrete and mortar are normally around 60–80% of the total volume of cementitious materials [47]. These values change with the formulation of printable material. Based on different studies, a certain interval value was highlighted for the mixture components such as cement, sand, mineral admixture, chemical admixture and Water to cement (W/C) ratio were proposed.

**Table 1.** Summary of the printability criteria (adapted from [23,29,40–43]).

| Criterion | Description | Control | Measurement |
|---|---|---|---|
| Flowability | Easy pumpability in the delivery, pumping system and easy deposition | Continuous grading of fine particles, use of powders additions (fly ash) and use of chemical additives (superplasticizers) | Direct measurement: rheological parameters by rheometers Indirect measurement: slump, slump flow, $T_{50}$ Slump and V-funnel |
| Extrudability | Ability of material to be delivered continuously through small pipes and nozzles without interruption | Fine particles, continuous grading, round shape and ratio nozzle diameter/maximal | Print test: the maximum length of printed filaments without blockage and without cracks. Ram extrusion |
| Buildability | Ability of the material to retain its shape just after deposition | Use of mineral additives, chemical additives (setting accelerator and rheology modifiers VMA (viscosity modifier agent)) | Assessment of mechanical properties at early age: penetration under constant load |
| Interlayer adhesion | Material ability to bond to subsequent layers. | Adjusted vertical print speed and use of admixtures (set retarder) | Hardened mechanical performance of inter-layers: direct tensile test and shear test |
| Open time | Defined as the time interval beyond which material extrudability property decrease not to confuse with setting time of the material. | As it is related to extrudability, Open time control been found by other researchers to varies using of powders additions, an increase in GGBS content, use of chemical additives that usually decreases cement setting time | Assessed with a Vicat apparatus to determine the initial and final setting time |

**Table 2.** Mix proportion of raw materials used for 3D printing.

| Research's Team | Cement (kg) | Silica Fume (kg) | Fly Ash (kg) | Sand (kg) | Water (kg) | W/C | Superplasticizer (% of Cement Weight) | Plasticizer (% of Cement Weight) | Retarder (% of Cement Weight) | Accelerator (% of Cement Weight) |
|---|---|---|---|---|---|---|---|---|---|---|
| B. Khoshnevis et al. [42] | 888 | _ | _ | 984 | 456 | 0.51 | 3 | _ | _ | _ |
| S. Lim et al. [23] | 579 | 83 | 165 | 1241 | 232 | 0.28 | 1 | 1.2 | 0.5 | _ |
| Z. Malaeb et al. [43] | 125 | _ | _ | 240 | 49 | 0.39 | _ | _ | 0.8 | 1.25 |
| B. Panda et al. [44] | 253 | 61 | 192 | 810 | 152 | 0.30 | 1.05 | _ | _ | _ |
| M. Krause et al. [45] | 430 | 180 | 170 | 1240 | 180 | 0.23 | 1.2 | _ | _ | _ |
| L. Wang et al. [31] | 1680 | 240 | 480 | _ | 270 | 0.27 | 1.08 | 1.2 | _ | _ |

## 2.2. Research Objectives: Experimental Framework for Printability

The objective of this paper was to propose a front-end experimental framework of printability pre-assessment of cementitious materials. This study examined the intrinsic properties of fresh mortar to introduce printability coefficients and their evolution during time and the 3D printing process. This front-end experimental framework comes to help in the first decisions of printability. The goal of this research was the development of a framework especially dedicated to laboratory testing that will foster 3D printing evolution and contribute to establish guidelines.

Figure 1 proposes a mix proportion interval based on the literature and a framework for printability assessment. The framework was based on practical experimentations with the support of literature reviews on formulating cementitious printable mortar [23,31,42–45]. The literature reviews exposed in the introduction highlights an interval value of mixture components such as cement, sand, mineral admixtures, chemical admixtures and water/cement ratio.

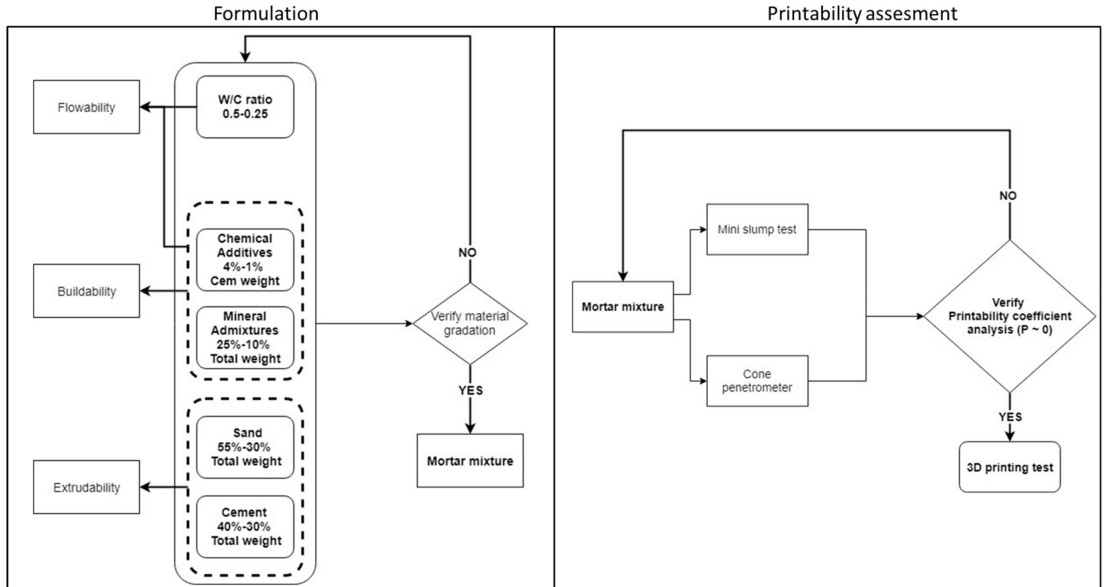

**Figure 1.** Proposed methodology for formulation and assessment of printability of cementitious material.

The first step of this framework was to use the mix-designs reported in the literature to define a mix-design to be investigated in this study. Each mixture was then tested for its intrinsic properties for assessment of printability. Three criteria were then influencing the formulation namely extrudability, flowability and buildability. Extrudability criterion was correlated to the grading and compaction, the use of fine particles, powder additions and chemical additions.

For this study, extrudability criterion was fulfilled by testing each formulation with a visual checking of extrusion for blockages. The extrusion was done using a universal screw pump with hose length of 5 m and a diameter of 27 mm.

Based on printability criteria presented in Table 1, two choices were made for this study, flowability, which represents how easy it is to pump in the delivery system and smooth deposition, and buildability, which represents the ability of the material to retain its shape just after deposition. These two criteria, flowability and buildability are then measured using respectively indirect measurement (slump test) and assessment of mechanical properties at an early age with a penetration under constant load. A printability coefficient is then introduced to correlate the flowability and buildability. The two criteria constitute a duality and contribute equally to printability. An open time interval is then introduced for the mixture under investigation.

## 3. Material and Method

### 3.1. Materials and Mix Design

In this study, the mixtures subjected to investigation are composed by cement, fine aggregate (Ø ≤ 3 mm), and water. The cement is an ordinary Portland cement (OPC) CEM I 52.5 N from Beaucaire. Tables 3 and 4 regroup different characteristic of the cement.

**Table 3.** CEM I 52.5 N chemical characteristic.

| Production Factory | Clinker ≥ 95% | | | $SO_3$ | $S^{2-}$ | $Na_2O$ Active Equivalent |
|---|---|---|---|---|---|---|
| | $C_3A$ | $C_3S$ | $C_2S$ | | | |
| **Beaucaire** | 2.3 | - | 10.2 | 2 | 0.01 | 0.32 |

**Table 4.** CEM I 52.5 N physical characteristic.

| Production Factory | Mechanical Resistance MPa | | | Finesse | | Cement Pure Paste | Beginning of Setting |
|---|---|---|---|---|---|---|---|
| | 1j | 2j | 28j | Blaine (cm²/g) | Refus (%) 40 µm | | |
| **Beaucaire** | 18 | 31 | 63 | 4280 | - | 28.6 | 3h10 |

The sand is a commercially available manufactured sand taken from a local supplier in the Nord of France. The grain size distribution of the sand is presented in Figure 2.

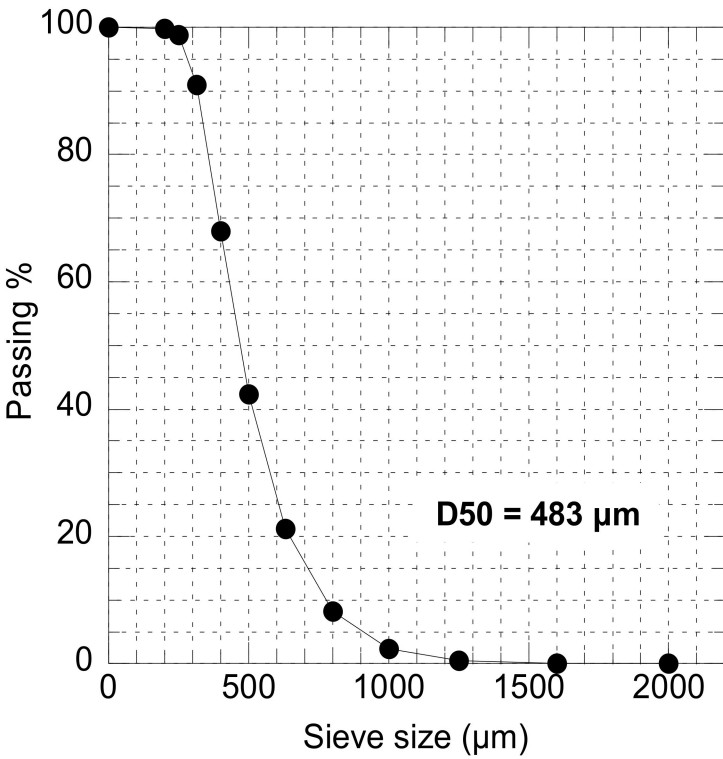

**Figure 2.** Distribution of sand grain.

Table 5 presents the chemical component and physical properties of the sand.

**Table 5.** Sand characteristic.

| | Chemical Element | Composition (%) | | Physical Properties | Value |
|---|---|---|---|---|---|
| **Chemical Characteristic** | SiO$_2$ | 99.30 | **Physical Property** | Bulk density | 2.65 |
| | Fe$_2$O$_3$ | 0.016 | | pH | 7 |
| | Al$_2$O$_3$ | 0.150 | | Apparent density | 1.6 |
| | TiO$_2$ | 0.017 | | Pyroscopic resistance | 1750 °C |
| | CaO | 0.006 | | CU (D60/D10) | 1.65 |
| | K$_2$O | 0.130 | | Fire loss | Max 0.20% |

Table 6 shows the mass proportions of three preliminary mixtures M1, M2 and M3. They were calculated assuming a cement's dosage of 35% of the total weight, a water to cement ratio of 0.36, 0.38 and 0.40, respectively. The temperature and relative humidity condition in the laboratory during mixing and measuring were in the range of 22–25 °C and 40–45%, respectively. The mixture was prepared in order to have a frictional flow when pumping.

**Table 6.** Mixture proportion for the trial.

| Mixture | Cement % | Sand % | Water % | W/C |
|---|---|---|---|---|
| M1 | 35 | 53 | 12 | 0.36 |
| M2 | 35 | 52 | 13 | 0.38 |
| M3 | 35 | 51 | 14 | 0.40 |

For the preparation of the mixtures, the dry materials were mixed at 2500 rpm for 2 min, the water was added and the whole was mixed for 8 min at the same speed. Finally, a final speed of 3500 rpm was performed for 2 min. The experiment began 10 min after the end of mixing and this moment was taken as reference of measurements (t = 0).

### 3.2. Printability Evaluation

The printability of a material represents its ability to be used successfully in a 3D printing process and it is essentially related to the mechanical and rheological properties at an early age [16]. The search for an optimal mixture for 3D printing necessitates several tests. In this study, the proposed parameters for deciding on the most appropriate solution have been identified as flowability, buildability and open time.

3.2.1. Material Distribution Flowability

The flowability represents the ease of pumpability and delivering system, the coefficient is dimensionless. It is based on the slump value and a reference maximum value of slump, which is the maximum value of slump for each mixture and a minimum reference.

The rheology of fresh mortar, even though very complex in nature, has been shown to follow the Bingham material model [48]. The rheology is directly related to 3D printing materials since the flow rate is reflected by pumping pressure which is dependent on yield stress [47,49]. Many mortar rheometers have been developed for measuring the rheological properties of fresh mortar using different principles and geometries [50]. However, they are generally expensive, complex to use and can give different absolute values [51]. As a result, there is a multitude of tests around the world to quantify or at least qualify the flowability of fresh mortar by measuring a quantity of geometry or time. Slump, slump flow, T50 slump, V-funnel and jump tables are the main tests used for printable materials [25]. The slump test still the cheaper, the simplest and shows a good correlation to the rheological properties. Therefore, the test is adopted in this study.

### 3.2.2. Shape Stability of Material Buildability

Buildability is another critical criterion, which refers to the ability of the material to retain its shape just after deposition, to be hard enough to support the weight of subsequent layers without collapsing and to bind to the next layers [25]. The criterion is therefore directly related to the initial mechanical stiffness.

The buildability represents the mechanical properties at an early age with a penetration under constant load, the coefficient is dimensionless. Buildability can be measured by the number of maximum layers that can hold or cylinder stability. Another study focuses on modeling of green mortar in terms of Young's modulus, Poisson's ratio, compressive and shear strength [14]. The penetration resistance method is also used to access the development of stiffness or the structural behavior of cementitious materials during the setting process. The precise distance impacted below the cone tip depends on the stiffness and thickness of the mortar paste being penetrated. Once the equilibrium of internal and external forces is reached, the applied load is resisted by equal stress resistance developed in the tested mortar. Therefore, the cone penetrometer is adapted and then adopted in this study.

### 3.2.3. Printability Windows (Open Time)

The open time of cementitious material is linked to the setting time, it is usually determined using a Vicat apparatus. However, this equipment is designed to assess the initial and final setting time, which does not give a characterizing change in workability during time. Some research investigations focus on monitoring the change of workability with time using a slump test [52,53]. This time window is the period during which the printing mixture could be extruded by the nozzle with acceptable quality in our case respond to flowability and buildability criteria considering the workability loss that occurs over time.

In our case, two-time limits are introduced to define the printability window of an optimal mixture and blockage time limits. The time $T_{optimal}$ is introduced to represent the optimal time to start printing after mixing, at this time the mixture responds to two of the criteria flowability and buildability. The second time (blockage limit) is the limit, which refers to the time when the loss of workability interferes with flowability criteria.

### *3.3. Tests and Procedures*

In this section, two experimental test methods used to characterize the printing mixtures and their results are presented. Firstly, conventional test methods and corresponding results are presented. These conventional tests serve as standard characterization of mixtures and provide the comparisons based for other studies.

### 3.3.1. Mini Slump Test

The slump test has been used successfully over decades to measure the workability of fresh mortar and their ability to flow under gravity. This method, where the test performing is according to NF EN 12350-2, is easy to handle in laboratories and on site mostly due to its simplicity, low cost and immediate results.

As presented in Figure 3, the apparatus consists of a mold, commonly named Abrams cone, in the shape of a truncated metal cone, open at both ends. In this study, the mini cone was used, and the geometries of these molds are given in Table 7:

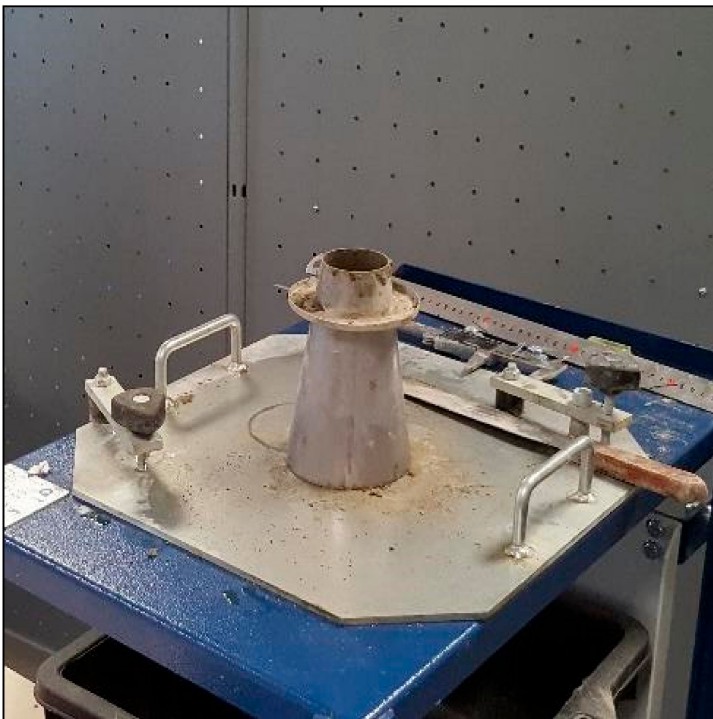

**Figure 3.** Mini slump test apparatus.

**Table 7.** Cone geometries difference.

| Parameter | Abrams Cone | Mini Cone |
|---|---|---|
| $H_0$ (mm) | 300 | 150 |
| $R_{min}$ (mm) | 100 | 50 |
| $R_{max}$ (mm) | 200 | 100 |

The test procedure is detailed in the following steps. The mold is placed centrally on the table and the mortar is introduced in three layers each layer compacted 25 times by a tamping rod in order to have a uniform filling. After that, excess mortar should be removed, and the surface leveled.

The decrease in height at the center point, known as slump S, is measured every 10 min with a total duration of 80 min, which corresponds to the typical duration of a 3D printing process. In this study, the dimensionless slump S' is also adopted, using the Equation (1):

$$S' = \frac{S}{H_0}. \tag{1}$$

- S': dimensionless slump;
- S: slump (mm);
- $H_0$: initial mini-slump height (mm).

### 3.3.2. Cone Penetrometer

The penetration resistance method was employed to assess the stiffness development or structural built up behavior of cementitious materials at the setting process [54]. The test used a cylindrical mold with a height of 35 mm and a diameter of 55 mm as presented in Figure 4.

The fresh cement mortar was firstly placed into the mold, then a steel nail: 35 mm long with a 30° angle and weighing 80 g, was released to penetrate into mortar paste under its own weight for 5 s. The digital readout provided readings in 0.1 mm resolution and the penetration depth was recorded every 10 min with a total duration of 80 min.

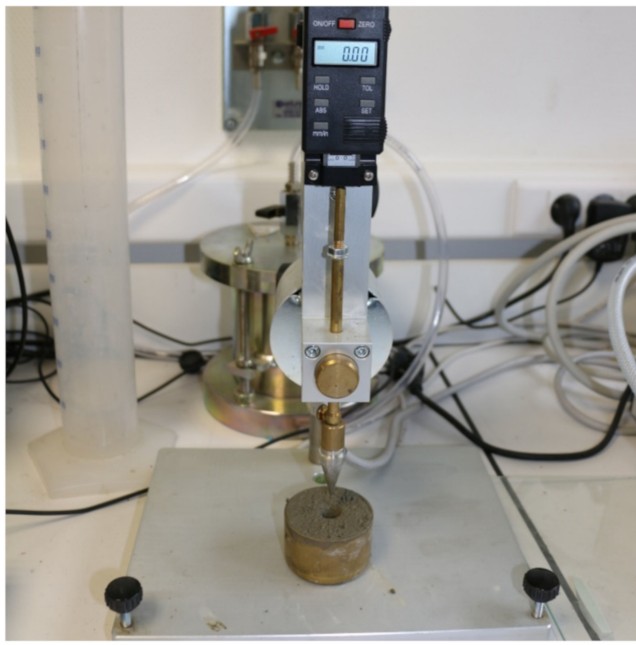

**Figure 4.** Cone penetrometer apparatus.

Usually the force penetration is measured for a constant imposed displacement [55]. As an alternative in this study, the depth was measured for a constant load and a qualitative stress penetration was introduced. Once the equilibrium of internal and external forces was reached, the applied load W was resisted by equal stress resistance developed in the tested mortar [56]. The penetration resistance defined as the normal stress on the side area of the used cone was calculated based on the Equation (2):

$$\sigma = \frac{W \cdot \sin(\theta)}{A_l} = \frac{m \cdot g}{\pi \cdot (1 + tan^2(\theta))} \cdot \frac{1}{h^2} \tag{2}$$

- $\sigma$: penetration resistance,
- $W$: applied load, (m mass),
- $\theta$: half the cone apex angle (15° for the used cone in this study),
- $A_l$: penetrated side area of the used cone in contact with mortar,
- $h$: penetration depth.

### 3.4. Mortar Printing System

A 3D mortar printing system based on extrusion has been developed in Ecole Centrale de Lille. The system was used to validate printable material property on a large-scale basis. Different components were linked together, to coordinate the task in hand. An industrial robotic arm was used to perform a variety of programmed tasks and trajectory for 3DCP (3D mortar printing) instead of the commonly used Cartesian robot (for Plastic 3D Printer) [16]. The printing speed was 100 mm/s (measure taken at the end of the nozzle) and the approximate pump flow rate was between 2.4 and 3 L/min. The mortar distribution to the system is a universal screw pump that is suitable for mortar distribution, the pump is connected to a nozzle and plays the role of the head printer. The head printer is composed of three components a camlock, which is connected to the tube of the pump, on the other side it is connected to a revolute mate connector and linked to an exchangeable piece of the nozzle. The passive head printer is a simple extrusion head, it has 400 mm in height and 15 mm of internal diameter. The nozzle work as a dispenser of mortar layer by layer and just follow the shape as given without any change in flow or extrusion control. The pump pressure was in a range of 0.3–6 Bar

(working pressure with mixture pumping), and the hose length and diameter were about 5 m and 27 mm, respectively [26].

The system was controlled and enslaved by an autonomous controller (PLC) for control and programming. Figure 5 presents the material and information flow of the complete experimental setup of a 3D mortar printer machine. Figure 6 shows experimental setup for 3D mortar printing used in this study for test validation.

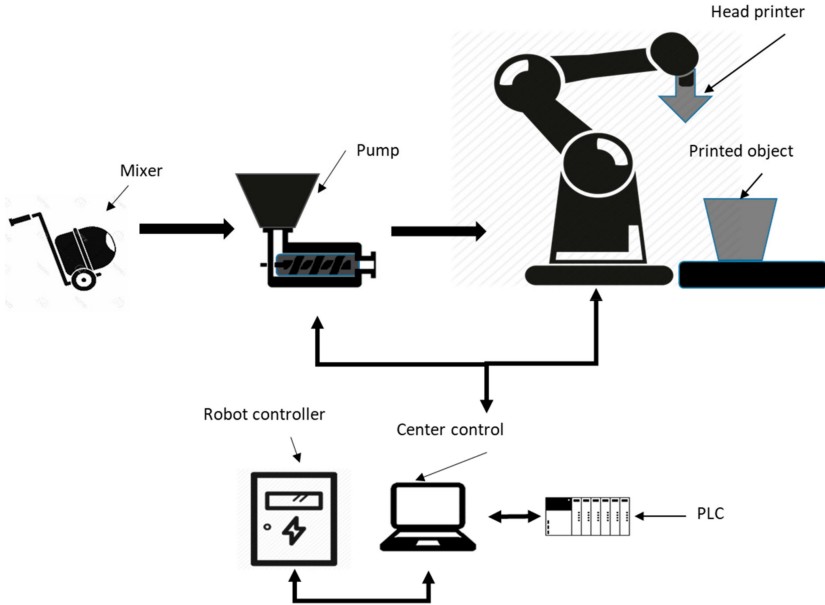

**Figure 5.** Complete experimental setup of 3D mortar printing.

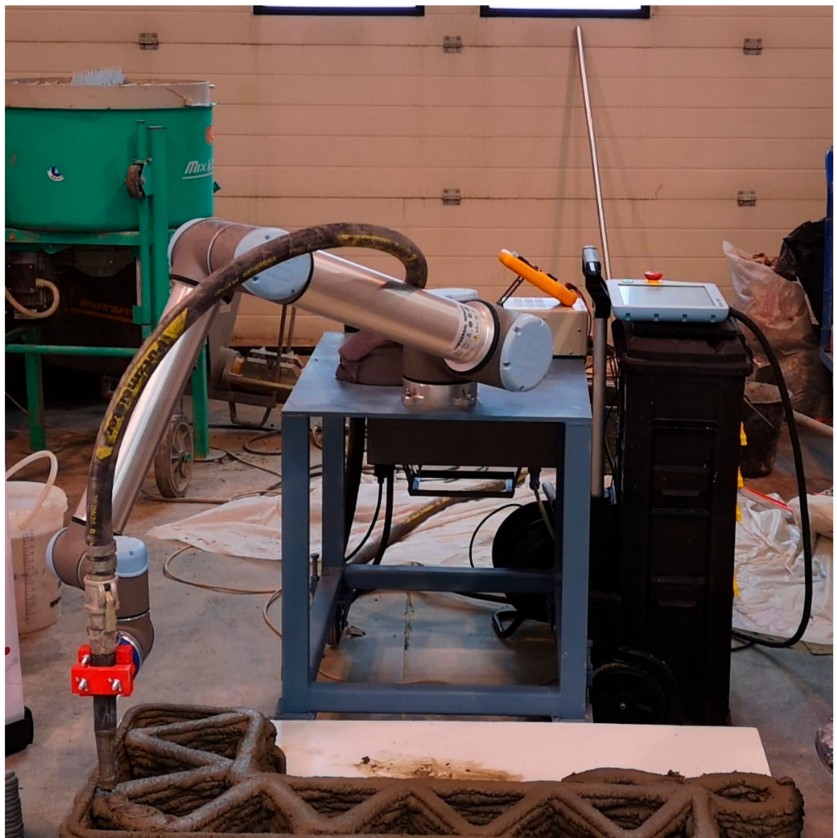

**Figure 6.** Experimental setup for 3D mortar printing.

## 4. Results and Discussion

### 4.1. Flowability Evaluation

Figure 7 illustrates the slump test as a dimensionless result of cementitious mixtures varying with rest time from t = 0 min to t = 80 min. As expected, the results show that the slump values of all mixes were decreasing over time. The phenomenon is known as 'slump loss', it occurs due to the reduction of mixing water, which is caused by aggregate absorption, evaporation and cement hydration [52]. The initial slump values of M1, M2 and M3 were 24 mm, 31 mm and 60 mm, respectively. The slump loss after 80 min was relatively the same for three mixes (33% for M1, M2 and 40% for M3). By referring to consistency classes and reasoning with dimensionless slump, M1 remains plastic and becomes firm between t = 40 min and t = 50 min. As for M2 and M3, they remained in the plastic range during the test period as illustrated in (d).

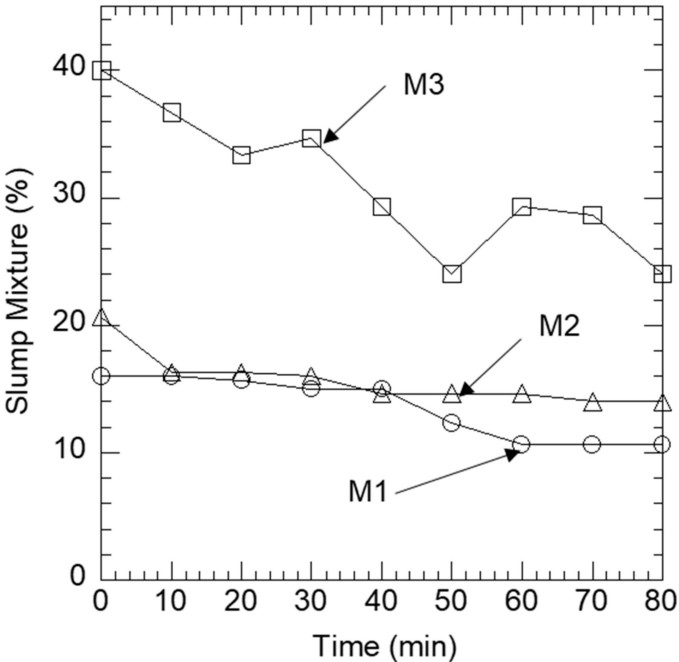

**Figure 7.** Result of the slump trial M1, M2 and M3.

### 4.2. Buildability Evaluation

Figure 8 illustrates the cone penetrometer results of cementitious mixtures varying with rest time from t = 0 min to t = 80 min and in terms of the penetration resistance. As expected, the penetration resistance has an increasing tendency since it is inversely proportional to the depth of penetration.

Literature authors have assumed that the rate of formation of Calcium silicate hydrate (CSH) crystal during the so-called "dormant" is constant because of the heat of hydration. Consequently, they conclude that, during this period, the evolution of the mechanical properties is linear then followed by an exponential part [57,58]. After 80 min, increases of resistance on order of 168%, 137% for M1, M2 while M3 remains relatively constant in terms of mechanical performance. The linear regression was adequate with coefficients of determination $R^2$ = 0.80, 0.91 and 0.97 for M1, M2 and M3, respectively. This regression shows that the method adopted in our study can be used to qualitatively evaluate the evolution of mechanical properties.

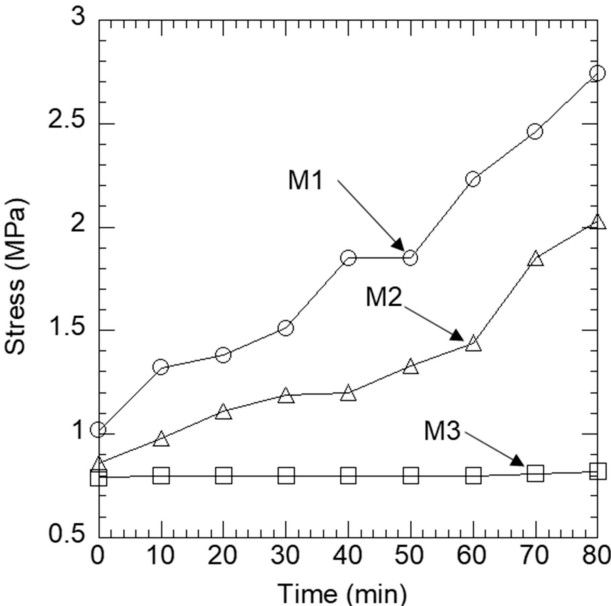

**Figure 8.** Penetration resistance results of trial mixtures M1, M2 and M3.

*4.3. Optimal Printability Coefficient*

From the previous results, we could conclude that M3 was the optimal formulation in terms of flowability and whereas M1 was the optimal formulation in terms of buildability, shape stability and mechanical stiffness. The two criteria constitute a duality and contribute equally to printability. Therefore, a simultaneous and dimensionless study of the two criteria must be conducted on the three formulations to decide on their printability. We introduced two coefficients F (Flowability) and B (Buildability) as specified below:

$$F = \frac{S - S_{min}}{S_{max} - S_{min}} \times 100. \tag{3}$$

- S: slump (mm);
- $S_{min}$: minimum slump of the mixture (mm);
- $S_{max}$: maximum slump of the mixture (mm).

$$B = \frac{\sigma - \sigma_{min}}{\sigma_{max} - \sigma_{min}} \times 100. \tag{4}$$

- $\sigma$: stress (MPa);
- $\sigma_{min}$: minimum stress of the mixture (MPa);
- $\sigma_{max}$: maximum stress of the mixture (MPa).

Figure 9 presents the two coefficients plotted against time for each mixture. M1 represents an equilibration between the two criteria that starts very early. From t = 40 min, the constructability criterion was already beginning to dominate. M2 seemed to be the best compromise. Indeed, the two criteria remained balanced for 50 min, which gives the largest window of printability (open time). However, M3 remained so flowable and the equilibration was expected to be too late, which is not interesting for 3D printing.

The two coefficients can be regrouped and analyzed to conclude about the printability, we introduced another coefficient called P (printability) as presented in the equation below

$$P(t) = F(t) - B(t). \tag{5}$$

- $F(t)$: flowability coefficient;

- B(t): buildability coefficient;
- The printability coefficient properties are presented as:
- P > 0: flowability property is predominant.
- P < 0: buildability property is predominant.
- P = 0: the two properties are balanced, and Figure 9 highlights an optimal time, Toptimal, which represents the optimal time to start printing (12 min after mixing).

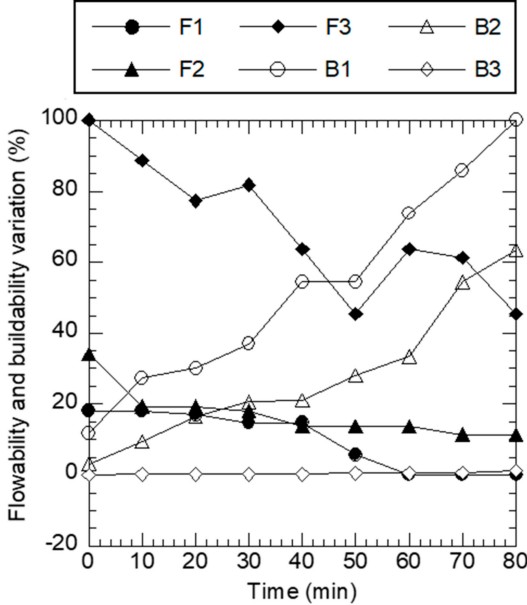

**Figure 9.** Flowability and buildability coefficients of trial mixtures M1, M2 and M3.

Figure 10 shows that the P2 coefficient, evolved from a P > 0 at the beginning of test measurement (0–12 min) and stays in the vicinity of P = 0 from 20 to 48 min, and decreased to a P < 0 where buildability is predominant. Indeed, M2 is the best compromise between the two criteria since it presents a large printability window. Thus, M2 is the optimal formulation that will be validated by performing a 3D printing test. The time interval, which the mixture is in the vicinity of P = 0, represents the open time interval of the actual batch, for this mixture the open time was 28 min.

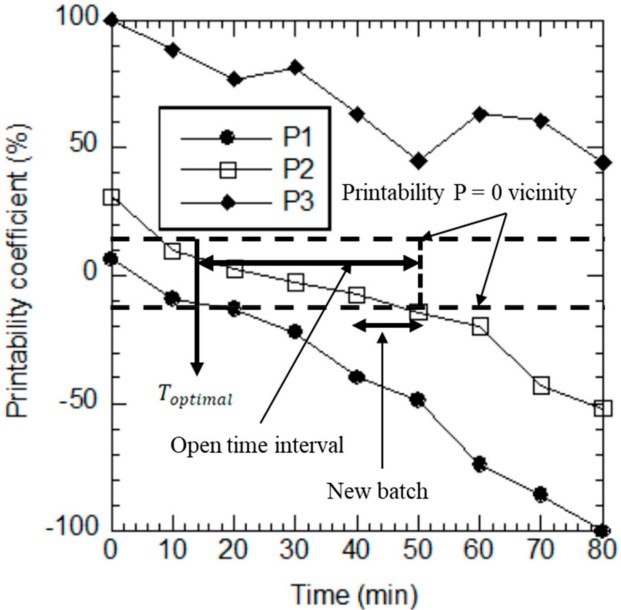

**Figure 10.** Printability coefficient evolution of trial mixtures.

## 5. 3D Printing Test of the Optimal Formulation

Based on the result from the above formulation, a large-scale test was performed. Full-scale laboratory production was necessary to demonstrate the application of the proposed criteria and mix design for their suitability in practice. In order to achieve this, a four-leaf clover, a golden rectangle and a four-leaf clover with torsion were printed with the optimal mixture M2.

The golden rectangle shows the possibilities of printing basic form with a visual effect on the texture of the printed element as displayed in Figure 11. The material is pasty and rough in the corner but soft in the upper face as illustrated by Figure 11. It does not present any cracks or fissuration at the base (first layer). The nozzle opening and geometry influence the layer thickness and buildability [59]. The printed component dimension was 350 mm height and 500 m width with 20 layers of 10 mm thickness.

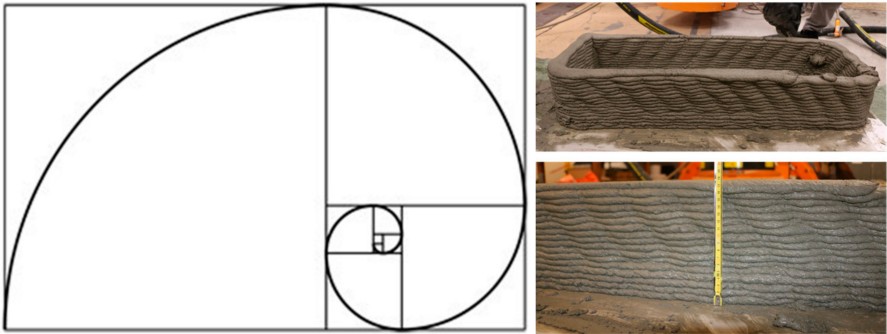

**Figure 11.** The golden rectangle model and printed part.

The four-leaf clover printed component was 240 mm height, 500 m width and 24 layers of 10 mm thickness, as illustrated in Figure 12. The trajectory has circular, sloped and direct path this demonstrates the minimum requirement of a mortar 3D printed material.

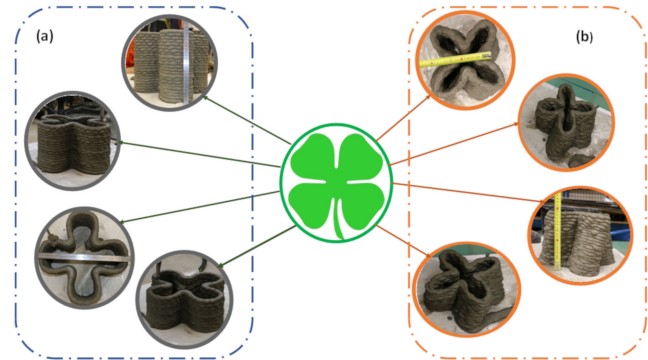

**Figure 12.** The four leaves (**a**) without torsion and (**b**) with torsion.

## 6. Conclusions

An experimental approach for printability assessment was developed. It consists of simple and easy-to-use tests and a dimensionless analysis to characterize the printability of cementitious material using specific indicators. The approach was applied to three formulations. The optimal mixture selected and successfully printed demonstrates the relevance in practice of the proposed criteria and the developed approach.

The water to cement ratio and cement's dosage of the total weight are two major parameters and the optimal values were demonstrated to be 0.38% and 35% respectively. The slump loss after 80 min was relatively the same for three mixtures 33% for M1, M2 and 40% for M3. In this time period of 80 min the increases of resistance of the mixes was on the order of 168%, 137% for M1, M2 while

M3 remained relatively constant in terms of mechanical performance. The mixture M2 was the best compromise between the two criteria since the P2 index stayed in the vicinity of P = 0 from 20 to 48 min. The evolution of the optimal formulation printability index P2 highlighted an optimal starting time to print, which was 12 min and started a new batch of mixture at 48 min.

The test for the optimal mixture on 3D printing machinery was successful with the mixture M2. The printed component dimension was 240 mm height and 500 m width with 24 layers of 10 mm thickness.

This paper opened multiple perspectives: continued the study of other formulations that contain powder additions and chemical additions in order to generalize and standardize the proposed model. Improved the model by working on a wide range of formulations in order to have a relevant reference values for slump and penetration as well as defining other printability indicators. Further research on buildability criteria based on rheological test and thixotropic model should be investigated. Buildability and flowability tests at various ambient conditions (temperature, humidity and wind velocity).

**Author Contributions:** Conceptualization, Z.L., A.Z.R. and N.Y.; methodology A.Z.R., S.e.M., and N.Y.; validation, Z.L., Z.D.; formal analysis, A.Z.R., S.e.M., and N.Y.; investigation, A.Z.R., S.e.M., and N.Y.; resources, Z.L.; writing—original draft preparation, A.Z.R., S.e.M., and N.Y.; writing—review and editing, Z.D., Z.L.; supervision, Z.L.; project administration, A.Z.R., S.e.M., and N.Y.; funding acquisition, Z.L.

**Funding:** This research was funded in the context of the industrial research chair "Construction 4.0".

**Conflicts of Interest:** The authors declare no conflict of interest.

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
