# Peer review of "Experimental Approach for Printability Assessment: Toward a Practical Decision-Making Framework of Printability for Cementitious Materials"

_buildings, doi:10.3390/buildings9120245_

Round 1
Reviewer 1 Report
Major Comments:
The title of this study is not matched with the conclusion. The title should be changed. How can we say "this is a practical-decision making framework" with this one study? Also, other relative studies follow this similar framework to test 3D printable cementitious mixtures. What is a clear difference between this study and other relative studies? All the literature reviews are related to 3D printing materials or histories of additive construction. Where are the literature reviews associated with the decision-making framework of 3D printability for cementitious materials?
The research objective is not clear because all the contents are not consistent from the beginning to the end. The reviewer was not able to figure out what the authors want to say. Is this study for the decision-making framework to determine whether the newly developed cementitious mixtures are appropriate for the additive construction? Or is this study for the 3D printable cementitious mixtures?
This manuscript involves many English grammar errors to be published in the journal. It supposed that the authors did not review this manuscript before they submit it.
Why was open time not investigated in this study? Open time is one of the main properties for the fresh properties of 3D printable cementitious mixtures. This study is incomplete about the rheology of fresh cementitious mixtures to say "this is a practical decision-making framework." To propose a practical decision-making framework, open time is more important than the “Large Scale 3D Printing Validation Test.” in this study.
Change all the words “concrete” to “mortar.”
Why the authors did not apply the “ASTM C1437 - 15 Standard Test Method for Flow of Hydraulic Cement Mortar” to investigate “flowability”? The material employed for this study is “mortar.” Coarse and fine aggregates were not used.
All the material properties used in this study should be specifically provided as Tables.
Figure 2 is not clear. Also, all the font and the font size in Figures must be consistent. All the qualities of the figures are very low to be published.
The purpose of the “Large Scale 3D Printing Validation Test” was not clear. It does not look like a large-scale model. What was the stacked height of the printed wall in Figure 10? It should be visible.
Minor Comments:
Line 5: “Front-end…” change to lowercase. Line 15: “Toptimal”?? What is “Toptimal”? Table 1: Please, relative information regarding open time. Line 110: “Front-end…” change to lowercase. Line 125: “Extrudability” change to lowercase. Line 126: “good grading” change to uppercase. Line 130: “Flowability” change to lowercase. Line 131: “Buildability” change to lowercase. Figure 1: Match the font with other figures. Line 143: “Grain” change to lowercase. Table 3: “Mixture” should be located at the center. Lines 182-184: “based on the resistance value and a reference maximum value of resistance which is the maximum value of resistance for each mixture and a minimum reference.” Check this sentence. Lines 190-191: “this stiffness and thickness are then interpreted in terms of qualitative stress.” Check this sentence.
Author Response
Dear reviewer,
Please see the attachment
Kind regards,

Reviewer 2 Report
The work is valuable because it presents practical tips for assessing the printability of cementitius materials. The authors proposed their own optimal printability factor. I suggest you make obvious corrections and additions:
1) Chapter 1. Introduction, text, row 32:I suggest adding: and unrecognized mechanical properties of the structure;
2) Chapter 3. Material and Method, text, row 144: no literature reference,
3) Chapter 3. Material and Method, table 1: what additives were used: plasticizer, retarder (as long as it's not a secret),
4) Chapter 3.3.1. Mini Slump, equ. 1: does m mean mass m = 80g,
5) Chapter 3.3.1. Mini Slump, Test, text row 243: was the speed measured at the inlet or at the end of the nozzle?
6) Chapter 3.3.1. Mini Slump, Test, text row 250: was the working pressure measured while the mixture was being fed?
7) Chapter 3.3.1. Mini Slump, Changing the length of the robot's working arm increased the volume of the mix. Interruptions in deliveries could cause unforeseen interruptions in work. How was the mass / volume controlled?
8) Chapter 4.3. Optimal Printability Coefficient, equ.4, No variable stresses in the formula
9) Chapter 5. Large Scale 3D Printing Validation Test, No detailed inspection of printed items. Have cramps. Have you determined the changes in deformation of the printed mortar strips?
10) Chapter 5. Large Scale 3D Printing Validation Test, What was the compressive and tensile strength of the mortar?
11) Chapter References: Duplicate literature numbering.
Author Response
Dear reviewer,
Please see the attachment.
Kind regards,

Reviewer 3 Report
The manuscript deals with an interesting topic that is appropriate for the Journal. A simple methodology for assessing the feasibility of a new laying technology of concrete is proposed.
However, some modifications of the manuscript are necessary before its acceptance for publication.
As a general comment, I have to remark that a better description of the materials investigated and of the testing procedures adopted is necessary. Particular attention should be devoted to the description of the methodologies developed or applied in this study, putting them in a separate section with respect to those developed and applied in the literature.
The numbering of the Figures should be checked. There are two Figures 3 (page 8 and page 9).-
Specific comments and suggestions are reported on the marked manuscript attached to this report.

Author Response

(The authors gave the same response as above.)

Round 2
Reviewer 1 Report
This manuscript is improved significantly.
Since the authors provided "open time", all studies associated with the fresh properties of the printing material are completed.
This reviewer believes that this manuscript is ready for publication.
This manuscript is a resubmission of an earlier submission. The following is a list of the peer review reports and author responses from that submission.
Round 1
Reviewer 1 Report
The paper entitled "Toward a Practical Decision-making Framework of Printability for Cementitious Materials" written by Zoubeir LAFHAJ, Andry Zaid RABENANTOANDRO, Soufiane EL MOUSSAOUI, Zakaria DAKHLI, and Nicolas YOUSSEF deals with an interesting subject and a case study. With poor explanations, details of the research are not realized. English writing is also poor i.e. too long sentences, spelling errors, repetition, and unacademic words. Major revision work is necessary to enhance the quality of the present manuscript.
The main points and remarks are as follows:
Abstract should be clear to a reader by including aims, the experimental procedure, results, discussion and conclusion briefly. Similarly, introduction should include a clear explanation of the main structure of article i.e. short introduction, literature and then experimental procedure Sequentially.Boston Consulting Office (BCN) confirms that large companies around the world are starting to use this technology and take advantage of its potential value. It is one of the pillars of Industry 4.0 since it can transform a virtual model into a real structure, which is already the case in the manufacturing industry: aeronautics, automotive, aerospace, medical and pharmaceutical. Today, the construction sector is being digitalized, an operation that has invaded the design phase and begins to reach the execution phase [2]. Due to its pros such as customized production, reduced waste, and diminished lead-times of prototyping object [3]. This front-end experimental framework comes to help in the first decisions of printability, this is a macro-test. Some feedback with researchers and professionals around the world has shown that there is a significant amount of time spent to find a decent formulation.
*These sections should be rewritten, because they aren’t clear enough and some references should be involved.
This technique offers valuable advantages: high-speed construction [7], removal of formwork [6], [8], less heavy labor [9], topological optimization and above all a great design flexibility [10][11]. In the past 35 years the development of new construction techniques based on 50 digitalization and automation flourished; such as implementing Contour crafting [14], D-shape [15], Freeform design construction [16], 3D concrete printing (TU Eindhoven) [5] and 3DConPrint [17]. Deposition of the concrete must mind the time gap between layer to avoid cold joints, there is a finite waiting time for the superposition of the new layers [23], [24].Most common criteria presented in the literature are extrudability, pumpability[17][25], buildability, open time, flowability [26] [27] and interlayer adhesion.*References should be involved at the end of a sentence as most of them refer to the same content.
Table 1 illustrates the different printable mixtures as shown in the literature.*Introduction should not include tables and figures.
The key challenges for printability were outlined by Wangler 68 et al. [21], main focus was the concrete extrusion and intermix with the previously deposited layer, it must support its own weight and the weight of the material to be subsequently deposited. Cementitious materials for extrusion-based, large-scale concrete printing has to exhibit paradox rheological properties to comply with required criteria [22].*Missing information of ref.21.
*sentences in this section are too long.
Some scientific studies are referred in the Experimental Framework for Printability without references. Please reconsider that. Conclusions are not conclusive, as just a summary of results. Paragraphs (1-5) are too detailed. It might be shortened. Highlights are missing.Author Response
Reviewer #1:
Abstract should be clear to a reader by including aims, the experimental procedure, results, discussion and conclusion briefly. Similarly, introduction should include a clear explanation of the main structure of article i.e. short introduction, literature and then experimental procedure Sequentially
An extensive change has been made to the abstract and introduction in order to meet the requirement.
These paragraphs were added:
Abstract
The objective of this paper is to propose a Front-end experimental framework of printability pre-assessment of cementitious materials. This study firstly presents a general review of additive manufacturing in construction and then examines the main characteristic of the material formulation and printability properties based on extrusion technique. This framework comes with experimental tests to determine a qualitative printability index of mixtures. It uses mix-designs reported in the literature to define interval ratio of mixture design to be investigated in this study. The focus was put on two criteria that influence the formulation namely flowability, buildability Two practiced based test mini slump and cone penetrometer was used to draw flowability and buildability dimensionless index. The results were analyzed by introducing an optimal printability coefficient and examining their time evolution. An optimal time of printing was determined Toptimal. Finally, a 3D concrete printing system and its operational process are presented. Then, based on the measurement, the optimal mixture is identified and printed in a large-scale geometry.
Introduction
This study will evaluate the materials performance of different printing trials. A front-end experimental framework of printability for cementitious materials is proposed long with the material and methodology used to validate the framework. It proposed an overview of literature review on practice-based test to give a first assessment of a printable material. It examines the intrinsic properties of fresh concrete to introduce printability coefficients and their evolution during time and the 3D printing process.
*These sections should be rewritten, because they aren’t clear enough, and some references should be involved.
This technique offers valuable advantages: high-speed construction [7], removal of formwork [6], [8], less heavy labor [9], topological optimization and above all a great design flexibility [10][11]. In the past 35 years the development of new construction techniques based on 50 digitalization and automation flourished; such as implementing Contour crafting [14], D-shape [15], Freeform design construction [16], 3D concrete printing (TU Eindhoven) [5] and 3DConPrint [17]. Deposition of the concrete must mind the time gap between layer to avoid cold joints, there is a finite waiting time for the superposition of the new layers [23], [24].Most common criteria presented in the literature are extrudability, pumpability[17][25], buildability, open time, flowability [26] [27] and interlayer adhesion.
These sections were rewritten with the change made through the introduction
This technique offers valuable advantages: high-speed construction, removal of formwork, less heavy labor, topological optimization and above all a great design flexibility [8]–[13]. Today, the construction sector is being digitalized, an operation that has invaded the design phase and begins to reach the execution phase [14]. A Boston Consulting Group (BCG) study in 2018 confirms that companies around the world are ready to use this technology and take advantage of its potential value [15]. However, this technology remains under-development, due to scaling up difficulties, absence of a business model in Europe, finished product quality and the intervention of several disciplines: design, architecture, materials science, computer and robotics [16].
In the past 20 years, the development of niche research on 3D printing in construction and automation flourished. In the international, here are a non-exhaustive name such as Contour Crafting and 3D-P (MIT) in United State, Winsun in China, D-Shape from Italy, CyBe Construction from the Netherlands, COBOD International from Danemark, Vertico from the Netherlands, Apis COR from Russia, XtreeE from France [17]–[19] . These niches have flourished but there are still major issues in the regulation of the material used in 3D printing for construction. In the European context various efforts have been made to facilitate introduction of new material linking normative agencies like Scientific and Technical Centre for Building (STCB) to startups or universities. The main country where the technology is sold for demonstrations are in the Middle East region (Saudi Arabia, Qatar, Dubai) due to it innovation policies.
*References should be involved at the end of a sentence as most of them refer to the same content.
Citation reference were modified accordingly
Table 1 illustrates the different printable mixtures as shown in the literature.
*Introduction should not include tables and figures.
A new section was added to the article for the table and figures that involved literature study
State of the art and research objectives
2.1 Literature Survey of 3D Printable Mixture
2.2 Research objectives: experimental framework for printability
The key challenges for printability were outlined by Wangler 68 et al. [21], main focus was the concrete extrusion and intermix with the previously deposited layer, it must support its own weight and the weight of the material to be subsequently deposited. Cementitious materials for extrusion-based, large-scale concrete printing has to exhibit paradox rheological properties to comply with required criteria [22].
*Missing information of ref.21.
*sentences in this section are too long.
The reference was controlled and reviewed for missing information.
Paragraph changed to:
However, it is commonly accepted that the material must respond to a certain number of specifications in order to be printable. The key challenges for printability were outlined by Wangler et al. also by Le et al. [12], [23]. The focus of those research was to assess the fresh properties of printable concrete. the concrete fresh properties of material using extrusion technique presented and assessed. Cementitious materials for extrusion-based concrete printing must exhibit paradox rheological properties to comply with the required criteria [24]. The printability of the material used is still the major challenge in the field [25].
Some scientific studies are referred in the Experimental Framework for Printability without references. Please reconsider that. Conclusions are not conclusive, as just a summary of results. Paragraphs (1-5) are too detailed. It might be shortened. Highlights are missing.
-Twelve other references along with the forty were added to strengthen the literature
-Those sentences were modified
Conclusion:
An experimental approach for printability assessment has been developed. It consists of simple and easy-to-use tests and a dimensionless analysis to characterize the printability of cementitious material using specific indicators. The approach was applied to three formulations. The optimal mixture selected and successfully printed demonstrates the relevance in practice of the proposed criteria and the developed approach.
The ratio water to cement and cement’s dosage of the total weight are two major parameters and the optimal values are demonstrated to be 0.38 and 35 % respectively. The slump loss after 80 min is relatively the same for three mixtures 33% for M1, M2 and 40 % for M3. In this time period of 80 min the increases of resistance of the mixes was on order of 168%, 137% for M1, M2 while M3 remains relatively constant in terms of mechanical performance. The mixture M2 was the best compromise between the two criteria since it P2 index stay in the vicinity of P= 0 from 20 minutes to 48 minutes. The evolution of the optimal formulation printability index P2 highlight an optimal starting time to print which was 12 minutes and start a new batch of mixture at 48 minutes.
The validation test on a 3D printing machinery was successful with the mixture M2. The printed component dimension was 240 mm height, 500 m width with 24 layers of 10 mm thickness.
This paper opens multiple perspectives: continue the study of other formulations that contain powder additions and chemical additions in order to generalize and standardize the proposed model. Improve the model by working on a wide range of formulations in order to have a relevant reference values for slump and penetration as well as defining other printability indicators. Further research on buildability criteria based on rheological test and thixotropic model should be investigated. Buildability, and flowability tests at various ambient conditions (temperature, humidity, wind velocity)
Reviewer 2 Report
In my opinion this paper should not be considered to publications in the journal. The authors did not contribute significantly for the progress of the research field. The research described in the manuscript was very simple and did not explore important questions reported in previous studies.
The manuscript proposes a methodology based in simple laboratory tests to develop printable cementitious materials. Three mixtures are proposed. They are composed only by Portland cement, sand and water. The only difference among the proposed mixtures is the water-to-cement ratio. The introduction of the paper does not reproduce the state of the art of the studied field. The authors could have explored many more references to propose a better research plan. For instance, by doing such deeper literature review they could have found many other reported studies discussing different methodologies to develop printable mix-designs. Furthermore, no information is given regarding the raw materials such as: sand grain size distribution and type of cement. In general, the report does not obey the common recommendations of a scientific writing. In many occasions the first person is used to state the opinion of the authors. Reaching the results part of the paper, the authors made very simple suggestions from the text and did not delivered important and reproducible methods. Some of the conclusions from the results are not in agreement with the findings of the study. Finally, in the conclusion, the results found are not taken in consideration and the authors arrived in conclusions that were not even approached in the report.
Additionally, it is important to recommend to the authors an English language revision on their manuscript.
Line 41: Review you citing methods. Instead of using [1], [2], [3], please use [1-3] or [1,3 and 4] in other cases.
Line 42: “ topological optimization” please cite [Domenico Asprone, Costantino Menna, Freek P. Bos, Theo A.M. Salet, Jaime Mata-Falcón, Walter Kaufmann, Rethinking reinforcement for digital fabrication with concrete, Cement and Concrete Research, Volume 112, 2018, Pages 111-121, ISSN 0008-8846, https://doi.org/10.1016/j.cemconres.2018.05.020. (http://www.sciencedirect.com/science/article/pii/S0008884618300309)].
Line 44: Please search for articles where sustainable printing mortars are also discussed
Line 45: “still in process of confidential research and development”. I do not agree with the authors statement. Indeed, there are several confidential researches going on, but there are a lot of papers being published recently on this subject.
Line 46: “absence of a business model”. I do not agree with the authors statement. There are companies selling equipment’s, materials and “know how”, such as these ones: Bruil, BESIX 3D and Vertico . Moreover, we should not forget that there are also contractors delivering the firsts printed infrastructure buildings such as [Theo A. M. Salet, Zeeshan Y. Ahmed, Freek P. Bos & Hans L. M. Laagland (2018) Design of a 3D printed concrete bridge by testing, Virtual and Physical Prototyping, 13:3, 222-236, DOI: 10.1080/17452759.2018.1476064] and plans for the delivery of other types of printed elements.
Line 47: please correct “science materials” to materials science.
Line 53: “Most of the process” please consider correct to “One of the most used techniques”.
Line 55: “tube” please correct to hoses.
Line 55: “head printer” please correct to printer head.
Line 58: “Materials printability still the major challenge [19].” Please consider changing to “The printability of the so-called printable materials is still the major challenge in the field”.
Line 58: “There is no relevant guideline for the 3DCP in terms of material formulation and no standard criteria are set to limit the specifications.” I disagree with the authors. There are several relevant papers where methodologies to develop and predict the printability of materials are proposed. Methodologies based on the mechanical performance of the fresh material or the study of the rheological properties based on experimental or numerical simulations were previously reported in literature.
Line 63: “inherent thixotropic property” this is usually attributed to mortars and not pastes.
Line 66: “Theses values changes” please consider correct it to “These values change”
Line 68-70: Separate this long sentence in at least two or more.
Line 71: “for extrusion-based, largescale concrete printing” please change to “for extrusion-based concrete printing”.
Line 73-74: “Deposition of the concrete must mind the time gap between layer to avoid cold joints, there is a finite waiting time for the superposition of the new layers.” I suggest to change to: “As previously reported in the literature, the interlayer adhesion between two printed layers is time sensitive.”
Line 77-78: “The Table 2 present certain of this criteria for printability such as its flowability, extrudability, buildability and interlayer adhesion.” I suggest to change to: “Table 2 presents cerain of this criteria for printability.”
Line 78-80: Please, I suggest to rephrase. Unfortunately, I could not understand what you were willing to say.
Line 82: “For that the article draws on the results of different printing campaign.” I suggest to change to: “This study will evaluate the materials performance of different printing trials”
Line 83: “Printability” please, always in the middle of the sentence do not use capital letters. They should be used only for proper noun. This remark is valid for the entire manuscript.
Table 1: What is a Superplastic? What is the difference of Superplastic to Plasticizer?
Table 2: Please do not forget to include in the Measurement column for Extrudability criteria the mixture assessment by ram extruder test usually reported for extrusion of ceramics and cementitious composites. This technique proved to be very efficient for the development of extrudable cementitious mixtures.
Table 2: In the column “control” for the buildability criteria, please do not forget to include the meaning of the abbreviation VMA.
Line 94: Delete “this is a macro test”.
Line 94: “Some feedback with researchers”, please if you have interviewed researches do not forget to cite those interviews. You can find instructions in: https://mdpi-res.com/data/mdpi_references_chicago_guide_v3.pdf
Line 95: Instead of “decent formulation” would be better to say “suitable formulation” or “the most appropriate mix-design”.
Line 95: I suggest to replace “The development of this kind of framework especially for laboratory testing will enable 3D printing evolution and guidelines.” to “The goal of this research is the development of a framework especially dedicated to laboratory testing that will foster 3D printing evolution and contribute to stablish guidelines.”
Line 97: Delete “The”.
Line 97: Replace “from” to “in”.
Line 98: Delete “different experimentation”
Line 98: Cite reference for: “The framework is based on literature reviews on formulating cementitious printable mortar”
Line 99: I guess that you were willing to say in “Literature reviews” the literature exposed in the introduction.
Line 100: Correct “chemical admixtures and W/C ratio.”
Line 100: Beginning of phrase with capital letter: “The”
Line 101: Replace “these returns of experiment” to “the mix-designs reported in the literature”
Line 101: Replace “choose an experimental mixture” to “to define a mix-design to be investigated in this study”.
Line 103: Replace “such as” to “namely:”
Line 105: Based in what parameters you arrived in this conclusion “For our study this criterion is fulfilled by a good grading and compaction for all the formulation”? Please, avoid the use of first person in scientific writing.
Line 107: Replace “article” to “study”
Line 107: Replace “the ease of pumpability” to “how easy to pump”
From here I decided to stop making language corrections. There are many more besides the ones that I have previously pointed out and further in the manuscript. Those mistakes make the reader’s understanding very difficult.
Figure 1: What do you mean with “verify material gradation”? If you mean the particle size distribution, your scheme is wrongly represented, as you have included the water in this assessment.
Figure 6: Keep only figure 6)d) as it summarizes all results in only one graph.
Figure 7: Keep only figure 7)d) as it summarizes all results in only one graph.
Figure 8: Keep only figure 8)d) as it summarizes all results in only one graph.
Line 291 – 293: I did not understand the reasons why the authors have decided that M2 is the best mixtures. They describe themselves that M1 had the best compromise between the evaluated criteria.
Line 304: As the nozzle cross section geometry and dimensions together with the layer height are important for the printing quality, they must be mentioned in the text.
Line 319: The data collected during the experimental study do not allow you to reach any conclusion regarding the optimum amount of Portland cement used in a printable mixture.
Author Response
Reviewer #2:
It is important to recommend to the authors an English language revision on their manuscript.
An English language review has been made to the manuscript
Review you citing methods. Instead of using [1], [2], [3], please use [1-3] or [1,3 and 4] in other cases.
Citation reference were modified accordingly
Line 42: “ topological optimization” please cite [Domenico Asprone, Costantino Menna, Freek P. Bos, Theo A.M. Salet, Jaime Mata-Falcón, Walter Kaufmann, Rethinking reinforcement for digital fabrication with concrete, Cement and Concrete Research, Volume 112, 2018, Pages 111-121, ISSN 0008-8846, https://doi.org/10.1016/j.cemconres.2018.05.020. (http://www.sciencedirect.com/science/article/pii/S0008884618300309)].
Line 44: Please search for articles where sustainable printing mortars are also discussed
Reference were added
Line 45: “still in process of confidential research and development”. I do not agree with the authors statement. Indeed, there are several confidential researches going on, but there are a lot of papers being published recently on this subject.
Line 46: “absence of a business model”. I do not agree with the authors statement. There are companies selling equipment’s, materials and “know how”, such as these ones: Bruil, BESIX 3D and Vertico . Moreover, we should not forget that there are also contractors delivering the firsts printed infrastructure buildings such as [Theo A. M. Salet, Zeeshan Y. Ahmed, Freek P. Bos & Hans L. M. Laagland (2018) Design of a 3D printed concrete bridge by testing, Virtual and Physical Prototyping, 13:3, 222-236, DOI: 10.1080/17452759.2018.1476064] and plans for the delivery of other types of printed elements.
A paragraph was added to explain our position about the business model.
In the past 20 years, the development of niche research on 3D printing in construction and automation flourished. In the international, here are a non-exhaustive name such as Contour Crafting and 3D-P (MIT) in United State, Winsun in China, D-Shape from Italy, CyBe Construction from the Netherlands, COBOD International from Danemark, Vertico from the Netherlands, Apis COR from Russia, XtreeE from France [17]–[19] . These niches have flourished but there are still major issues in the regulation of the material used in 3D printing for construction. In the European context various efforts have been made to facilitate introduction of new material linking normative agencies like Scientific and Technical Centre for Building (STCB) to startups or universities. The main country where the technology is sold for demonstrations are in the Middle East region (Saudi Arabia, Qatar, Dubai) due to it innovation policies.
Line 47: please correct “science materials” to materials science.
Line 53: “Most of the process” please consider correct to “One of the most used techniques”.
Line 55: “tube” please correct to hoses.
Line 55: “head printer” please correct to printer head.
Line 58: “Materials printability still the major challenge [19].” Please consider changing to “The printability of the so-called printable materials is still the major challenge in the field”.
Lines corrected
Line 58: “There is no relevant guideline for the 3DCP in terms of material formulation and no standard criteria are set to limit the specifications.” I disagree with the authors. There are several relevant papers where methodologies to develop and predict the printability of materials are proposed. Methodologies based on the mechanical performance of the fresh material or the study of the rheological properties based on experimental or numerical simulations were previously reported in literature.
There are several papers where methodologies are proposed but none are approved as a standard for normative purpose, those proposal are still under refinement and development.
Line 63: “inherent thixotropic property” this is usually attributed to mortars and not pastes.
Line 66: “Theses values changes” please consider correct it to “These values change”
Line 68-70: Separate this long sentence in at least two or more.
Line 71: “for extrusion-based, largescale concrete printing” please change to “for extrusion-based concrete printing”.
Line 73-74: “Deposition of the concrete must mind the time gap between layer to avoid cold joints, there is a finite waiting time for the superposition of the new layers.” I suggest to change to: “As previously reported in the literature, the interlayer adhesion between two printed layers is time sensitive.”
Line 77-78: “The Table 2 present certain of this criteria for printability such as its flowability, extrudability, buildability and interlayer adhesion.” I suggest to change to: “Table 2 presents cerain of this criteria for printability.”
Line 78-80: Please, I suggest to rephrase. Unfortunately, I could not understand what you were willing to say.
Line 70-80 were modified and rephrased to be clearer
The printability of a possible printable material must respond to these criteria extrudability, buildability, open time, flowability / workability / pumpability and interlayer adhesion. Each criterion is described and presented with it control dispositive. A proposed measurement test is presented that regroup different literature survey [23], [29], [37]–[40].
Table 2 illustrates the different printable mixtures as shown in the literature. Portland cement (PC) is the prime ingredient binder in most of the formulations presented through the literature due to its inherent thixotropic property [41]. For example, the ratio of water to cement occupies a wide range of values from 0.27 to 0.51 and no limitations can be concluded. The aggregates volume for cementitious material such as concrete and mortar are normally around 60% to 80% of the total volume of cementitious materials [42]. These values change with the formulation of printable material. Based on different studies an interval value of mixture components such as cement, sand, mineral admixture, chemical admixture, W/C ratio were proposed
Line 82: “For that the article draws on the results of different printing campaign.” I suggest to change to: “This study will evaluate the materials performance of different printing trials”
Line 83: “Printability” please, always in the middle of the sentence do not use capital letters. They should be used only for proper noun. This remark is valid for the entire manuscript.
The manuscript was review for capital letters errors
Table 1: What is a Superplastic? What is the difference of Superplastic to Plasticizer?
It is a superplasticizer not superplastic, for the difference in terms of effect on the concrete batch the plasticizer improves the workability and reduce slightly the water/cement ratio. Superplasticizer is a water-reducing admixture capable of producing large water reduction or great flowability without causing undue set retardation or entrainment of air in mortar or concrete
Table 2: Please do not forget to include in the Measurement column for Extrudability criteria the mixture assessment by ram extruder test usually reported for extrusion of ceramics and cementitious composites. This technique proved to be very efficient for the development of extrudable cementitious mixtures.
Ram extrusion was added to the column
Table 2: In the column “control” for the buildability criteria, please do not forget to include the meaning of the abbreviation VMA.
VMA (Viscosity Modifier Agent)). Was added
Line 94: Delete “this is a macro test”.
Line 94: “Some feedback with researchers”, please if you have interviewed researches do not forget to cite those interviews. You can find instructions in: https://mdpi-res.com/data/mdpi_references_chicago_guide_v3.pdf
Line 95: Instead of “decent formulation” would be better to say “suitable formulation” or “the most appropriate mix-design”.
Line 95: I suggest to replace “The development of this kind of framework especially for laboratory testing will enable 3D printing evolution and guidelines.” to “The goal of this research is the development of a framework especially dedicated to laboratory testing that will foster 3D printing evolution and contribute to stablish guidelines.”
The paragraph was changed to:
A qualitative printability evaluation was set, and the formulations compared according to those indicators. The development of this kind of framework especially dedicated to laboratory testing will foster 3D printing evolution and contribute to establish guidelines.
Line 97: Delete “The”.
Line 97: Replace “from” to “in”.
Line 98: Delete “different experimentation”
Line 98: Cite reference for: “The framework is based on literature reviews on formulating cementitious printable mortar”
Line 99: I guess that you were willing to say in “Literature reviews” the literature exposed in the introduction.
Line 100: Correct “chemical admixtures and W/C ratio.”
Line 100: Beginning of phrase with capital letter: “The”
Line 101: Replace “these returns of experiment” to “the mix-designs reported in the literature”
Line 101: Replace “choose an experimental mixture” to “to define a mix-design to be investigated in this study”.
Line 103: Replace “such as” to “namely:”
Line 105: Based in what parameters you arrived in this conclusion “For our study this criterion is fulfilled by a good grading and compaction for all the formulation”? Please, avoid the use of first person in scientific writing.
Line 107: Replace “article” to “study”
All those grammars were taken into account by the authors.
Line 107: Replace “the ease of pumpability” to “how easy to pump”
Line 97-107 were correct accordingly
Figure 1: What do you mean with “verify material gradation”? If you mean the particle size distribution, your scheme is wrongly represented, as you have included the water in this assessment.
Verify material gradation here refers to the maximum particle size accepted by the distribution system for this case study it is 3 mm.
Figure 6: Keep only figure 6)d) as it summarizes all results in only one graph.
Figure 7: Keep only figure 7)d) as it summarizes all results in only one graph.
Figure 8: Keep only figure 8)d) as it summarizes all results in only one graph.
Change has been made to reflect the review
Line 291 – 293: I did not understand the reasons why the authors have decided that M2 is the best mixtures. They describe themselves that M1 had the best compromise between the evaluated criteria.
The optimal mixture was M2 not M1 because the P2 coefficient, evolve from a P > 0 at the beginning of test measurement (0 to 12 minutes) and stay in the vicinity of P= 0 from 20 minutes to 48 minutes, and decreased to a P < 0 where buildability is predominant. Indeed, M2 is the best compromise between the two criteria since it presents a large printability window.
Change has been made to correct this
Line 304: As the nozzle cross section geometry and dimensions together with the layer height are important for the printing quality, they must be mentioned in the text.
A sentence on this was added to the 3D printing validation section.
Line 319: The data collected during the experimental study do not allow you to reach any conclusion regarding the optimum amount of Portland cement used in a printable mixture.
The data collect does not allow to reach conclusion on the optimum amount of PC. For a different mixture with another material composition the optimum PC value would change. In this case study with the material that we used to test the framework; the optimum amount of PC was 35%. For another composition it may change to another amount based on the utilization of the framework.
Reviewer 3 Report
Major Comments:
Authors should review the entire manuscript slowly to improve the quality of this manuscript because there are many typos and expressions are not consistent. Even, authors used “Buildings” template to submit this manuscript to “Materials”. Since this is not a conference processing, please review the entire manuscript multiple times.
In this field (Digital Construction or Additive Construction), opinions are divided over whether we can use “concrete” because they don’t use coarse aggregates. I recommend revising the word “concrete” to “mortar”.
What is the novelty of this study? Why you tested it? What we did not yet know? Please, briefly and clearly add the problem statement and the objective of this study at the end of the introduction.
Section 3.1 Materials and Mix Design is not enough. Please, provide all the material properties employed for this study specifically as tables.
Line 171: What the clear differences between slump test (NF EN 12350-2) and slump test (ASTM)? Also, please, add its reference.
(1) and Eq. (2): Where did you get these equations? Please, add their references.
Flowability is very important compared to printabiltiy, buildability, and etc. Can you add the extra explanations for 4.1 Flowability Evaluation?
Lines 252-255: How many data did you get for linear regression analysis regarding M1, M2, and M3? Did linear regression models satisfy with the central limit theorem? Can we trust the coefficients of determinations?
Thee formulations should be emphasized with additional explanations. The explanations are not enough.
The conclusion is not enough to sum up the main points of this study. Please, explain what the authors figured out through this study in detail.
Minor Comments:
Line 13-14: Uppercase and lowercase letters used at the same time. Please, make them consistently.
Line 21: Typo. It should be “Analyzed”.
Line 30: Please, add the ‘definite article’ or the ‘indefinite article’ in front of “process”.
Line 50: Please, add a comma after ‘35 years’.
Line 53: “are” or “is”?
Line 63: Please add ‘of’ after “the ratio”. Or you can write the water/cement ratio.
Line 65: Check the location of a comma after “concrete”.
Line 69: Please, add the ‘definite article’ or the ‘indefinite article’ in front of “process”.
Line 72: Please, add the ‘definite article’ or the ‘indefinite article’ in front of “required”.
Line 77: Remove “The”.
Line 77: Please, check which one is right between “this criterion” and “these criteria”.
Line 81: Please, add a comma after ‘this study’.
Line 82: Please, add a comma after ‘for that’.
Line 83: “were” or “was”.
Line 83: “P” should be lowercase letter. “printabilty”
Tables 1 and 2: The format must be the directions of MDPI.
From Page 4: Authors used the template of “Buildings”. Not “Materials”.
Line 92: Please, add the ‘definite article’ or the ‘indefinite article’ in front of “time”.
Line 97: Remove “The”.
Line 101: Please, add the ‘definite article’ or the ‘indefinite article’ in front of “experiment”.
Line 105: Please, add a comma after ‘study’.
Line 119: “of” or “by”?
Line 128: The formatting of Table 3 should be checked.
Line 159: It should be “Modeling”.
Line 178: The formatting of Table 4 should be checked.
Line 191: Please, add “in” between “presented” and “Figure 3”.
Line 211: “in” or ”on”?
Line 217: Add “the” in front of “head”.
Line 217: “by” or ”of”?
Line 219: Add “the” in front of “nozzle”.
Line 225: Remove “The” in front of “Figure 4”.
Line 225: It should be “presents”.
Line 235: It should be “is known as”.
Line 239: Please, add the ‘definite article’ or the ‘indefinite article’ in front of “dimensionless”.
Line 249: It should be “because of”.
Line 271: Remove ”The” in front of “Figure 8”.
Line 280: It should be “analyze”.
Line 287: Remove ”The” in front of “Figure 9”.
Line287: It should be “highlights”.
Line 289: Add “the” in front of “P2”.
Line 318: Add “the” in front of “total”.
Line 417: Please, check reference.
Reference List: Please, add the access dates and doi for all the journal articles for crosschecking.
Author Response
Major Comments:
Authors should review the entire manuscript slowly to improve the quality of this manuscript because there are many typos and expressions are not consistent. Even, authors used “Buildings” template to submit this manuscript to “Materials”. Since this is not a conference processing, please review the entire manuscript multiple times.
An extensive change has been made to the template, abstract and introduction in order to meet the requirement.
What is the novelty of this study? Why you tested it? What we did not yet know? Please, briefly and clearly add the problem statement and the objective of this study at the end of the introduction
This study firstly presents a general review of additive manufacturing in construction and then examines the main characteristic of the material formulation and printability properties based on extrusion technique. This framework comes with experimental tests to determine a qualitative printability index of mixtures. It uses mix-designs reported in the literature to define interval ratio of mixture design to be investigated in this study. The focus was put on two criteria that influence the formulation namely flowability, buildability Two practiced based test mini slump and cone penetrometer was used to draw flowability and buildability dimensionless index. The results were analyzed by introducing an optimal printability coefficient and examining their time evolution.
Section 3.1 Materials and Mix Design is not enough. Please, provide all the material properties employed for this study specifically as tables.
Chemical characteristic and grain size distribution of the material used were added to the manuscript
This was not added before because the framework is used to make abstraction of those properties for a first formulation
Line 171: What the clear differences between slump test (NF EN 12350-2) and slump test (ASTM)? Also, please, add its reference.
The difference between the two standards are:
the American standards explicitly state the dimension of the cone and the lifting procedure must be done without any rotational movement at all (Lyons, Arthur (2007). Materials for architects and builders. Butterworth-Heinemann. Retrieved 2010-12-11).
The European standard specify that the cone should only be lifted vertically and the of the slump cone must be done in three equal layers with the mixture being tamped down 25 times each layer (Tattersall, G.H. (1991). Workability and quality control of concrete. London: E & FN Spon. ISBN 0-419- 14860-4)
(1) and Eq. (2): Where did you get these equations? Please, add their references.
The equation (1) was determined as a dimensionless number in order to be independent of scale. Hence, it can be easily used for different scale-analogue problems and interpretation without geometrical issues.
The equation (2) was determine through constraint-based formula with the geometry of the cone.
Lines 252-255: How many data did you get for linear regression analysis regarding M1, M2, and M3? Did linear regression models satisfy with the central limit theorem? Can we trust the coefficients of determinations?
Those tests were done with different kind of sand with different grain size distribution. Two of the mixture batch did not pass the material gradation phase and could not be extrude. The last three formulation was repeated three times each.
The conclusion is not enough to sum up the main points of this study. Please, explain what the authors figured out through this study in detail.
The conclusion was modified and detailed for the study
An experimental approach for printability assessment has been developed. It consists of simple and easy-to-use tests and a dimensionless analysis to characterize the printability of cementitious material using specific indicators. The approach was applied to three formulations. The optimal mixture selected and successfully printed demonstrates the relevance in practice of the proposed criteria and the developed approach.
The ratio water to cement and cement’s dosage of the total weight are two major parameters and the optimal values are demonstrated to be 0.38 and 35 % respectively. The slump loss after 80 min is relatively the same for three mixtures 33% for M1, M2 and 40 % for M3. In this time period of 80 min the increases of resistance of the mixes was on order of 168%, 137% for M1, M2 while M3 remains relatively constant in terms of mechanical performance. The mixture M2 was the best compromise between the two criteria since it P2 index stay in the vicinity of P= 0 from 20 minutes to 48 minutes. The evolution of the optimal formulation printability index P2 highlight an optimal starting time to print which was 12 minutes and start a new batch of mixture at 48 minutes.
The validation test on a 3D printing machinery was successful with the mixture M2. The printed component dimension was 240 mm height, 500 m width with 24 layers of 10 mm thickness.
This paper opens multiple perspectives: continue the study of other formulations that contain powder additions and chemical additions in order to generalize and standardize the proposed model. Improve the model by working on a wide range of formulations in order to have a relevant reference values for slump and penetration as well as defining other printability indicators. Further research on buildability criteria based on rheological test and thixotropic model should be investigated. Buildability, and flowability tests at various ambient conditions (temperature, humidity, wind velocity).
Reference List: Please, add the access dates and doi for all the journal articles for crosschecking.
The DOI were added to the reference
Minor Comments:
The minors comment were corrected accordingly
Thank you for your consideration, I look forward to hearing from you,
Round 2
Reviewer 3 Report
The initial manuscript was too hard to understand. But this manuscript has improved significantly. I can understand what the authors are trying to say. The current manuscript is fine for publication.